# Anti-Apolipoprotein A-1 IgG Influences Neutrophil Extracellular Trap Content at Distinct Regions of Human Carotid Plaques

**DOI:** 10.3390/ijms21207721

**Published:** 2020-10-19

**Authors:** Rafaela F. da Silva, Daniela Baptista, Aline Roth, Kapka Miteva, Fabienne Burger, Nicolas Vuilleumier, Federico Carbone, Fabrizio Montecucco, François Mach, Karim J. Brandt

**Affiliations:** 1Division of Cardiology, Foundation for Medical Researches, Department of Medicine Specialties, Faculty of Medicine, University of Geneva, Av. de la Roseraie 64, 1211 Geneva, Switzerland; rfdasilva.ufmg@gmail.com (R.F.d.S.); daniela.baptista@unige.ch (D.B.); aline.roth@unige.ch (A.R.); kapka.miteva@unige.ch (K.M.); fabienne.burger@unige.ch (F.B.); francois.mach@unige.ch (F.M.); 2Department of Physiology and Biophysics, Institute of Biological Sciences, Federal University of Minas Gerais, 31270-901 Belo Horizonte, Brazil; 3Department of Diagnostics, Division of Laboratory Medicine, Geneva University Hospitals, 1211 Geneva, Switzerland; nicolas.vuilleumier@hcuge.ch; 4Department of Medical Specialities, Division of Laboratory Medicine, Faculty of Medicine, 1211 Geneva, Switzerland; 5First Clinic of Internal Medicine, Department of Internal Medicine, University of Genoa, viale Benedetto XV n6, 16132 Genoa, Italy; federico.carbone@edu.unige.it (F.C.); fabrizio.montecucco@unige.it (F.M.); 6IRCCS Ospedale Policlinico San Martino Genoa-Italian Cardiovascular Network, Largo Rosanna Benzi n10, 16132 Genoa, Italy

**Keywords:** atherosclerosis, vulnerable plaques, anti-apoA-1 IgG, neutrophil extracellular traps

## Abstract

Background: Neutrophils accumulate in atherosclerotic plaques. Neutrophil extracellular traps (NET) were recently identified in experimental atherosclerosis and in complex human lesions. However, not much is known about the NET marker citrullinated histone-3 (H3Cit) expression and functionality in human carotid plaques. Moreover, the association between the proatherosclerotic autoantibody anti-apolipoprotein A-1 (anti-ApoA-1 IgG) and NET has never been investigated. Methods: Atherosclerotic plaques have been obtained from 36 patients with severe carotid stenosis that underwent carotid endarterectomy for severe carotid stenosis. Samples were sectioned into upstream and downstream regions from the same artery segment. Plaque composition and expression of NET markers neutrophil elastase (NE) and H3Cit were quantified by immunohistochemistry. H3Cit expression and function was evaluated by immunofluorescence and confocal analysis in a subset of patients. Results: Pathological features of vulnerable phenotypes were exacerbated in plaques developed at downstream regions, including higher accumulation of neutrophils and enhanced expression of NE and H3Cit, as compared to plaques from upstream regions. The H3Cit signal was also more intense in downstream regions, with significant extracellular distribution in spaces outside of neutrophils. The percentage of H3Cit colocalization with CD66b (neutrophils) was markedly lower in downstream portions of carotid plaques, confirming the extrusion of NET in this region. In agreement, the maximum distance of the H3Cit signal from neutrophils, extrapolated from vortex distance calculation in all possible directions, was also higher in downstream plaques. The serum anti-ApoA-1index positively correlated with the expression of H3Cit in downstream segments of plaques. Expression of the H3Cit signal outside of neutrophils and H3Cit maximal distance from CD66b-positive cells increased in plaques from serum positive anti-ApoA-1 patients compared with serum negative patients. Conclusion: NET elements are differentially expressed in upstream versus downstream regions of human carotid plaques and may be influenced by circulating levels of anti-ApoA-1 IgG. These findings could warrant the investigation of NET elements as potential markers of vulnerability.

## 1. Introduction

Atherosclerosis is a chronic inflammatory disease with an immune-driven component [1]. Indeed, atherosclerotic lesions are infiltrated by T-cells and antibodies directed against diverse autoantigens. Neutrophils accumulate in atherosclerotic plaques induced by Western diet in distinct experimental mouse models of the disease [2]. In complex human plaques, neutrophils are present in the fibrous cap or in the shoulder, in the interface to media, or in areas with intraplaque haemorrhaging [3]. Previous works from our group showed in a large cohort study that human carotid plaques that develop in upstream blood flow regions present a distinct composition compared to plaques downstream [4]. Indeed, extensive histological analysis of this large cohort showed that plaques from downstream regions presented several features of a more vulnerable phenotype.

In 2004, Brinkmann et al. were the first to show that activated neutrophils are able to release net-like structures, which were able to kill bacteria extracellularly [5]. These structures consisted of chromatin DNA strands coated with histones and granular proteases and were named neutrophil extracellular traps (NETs). The mechanism by which neutrophils produce and release NETs is complex and it varies upon stimuli, yet two major pathways have been described according to their dependence on nicotinamide adenine dinucleotide phosphate (NADPH) oxidase [6]. The NADPH oxidase-dependent mechanism involves the production of reactive oxygen species (ROS); leakage of multiple proteinases, such as neutrophil elastase (NE), proteinase 3, cathepsin G, gelatinase, and the pro-oxidant enzyme myeloperoxidase (MPO); and citrullination of nuclear histones [7,8]. The NADPH oxidase-independent mechanism is less understood, yet in vitro studies from Kenny et al. have demonstrated that diverse stimuli have distinct ROS requirement for NETosis [9]. Besides its participation in the host defense against pathogens, the release of NET can also be involved in the onset and/or progression of several pathological conditions [10,11]. In the context of atherosclerosis, NETs can be induced by cholesterol crystal in vitro [12], and certain NET elements have been identified in atherosclerotic plaques developed in distinct experimental mouse models. In humans, NET elements were identified in the luminal area of carotid plaques and in distinct types of coronary atherothrombosis [13,14,15].

The presence of serum autoantibodies against apolipoprotein A-1 (ApoA-1), the major protein fraction of HDL, is considered as an independent predictor of cardiovascular events, and an active modulator of human atherothrombosis [16]. Recent studies from our group have shown that elevated serum levels of ApoA-1 antibodies (anti-apoA-1 IgG) can predict unfavourable cardiovascular events in patients with myocardial infarction (MI) [17,18,19], severe carotid stenosis [20,21], rheumatoid arthritis (RA) [22], and also in the general population [23]. Moreover, the presence of serum anti-ApoA-1 IgG was associated with increased histological features of atherosclerotic plaque vulnerability in human biopsies of carotid arteries [24] and in mice studies. It indicates that passive immunization of apolipoprotein E knockout (Apoe^−/−^) mice with human anti-apoA-1 IgG promotes atherogenesis and atherosclerotic plaque vulnerability [25].

In the present study, we investigated the expression of NET elements in human carotid plaques from upstream and downstream regions of the maximum stenosis, known to be exposed to different hemodynamic forces. High wall shear stress characterizes the upstream region while the downstream region is associated with low wall shear stress and turbulent blood flow [4]. Due to previous associations demonstrated between serum anti-ApoA-1 IgG positivity and neutrophil accumulation in atherogenesis [25], we evaluated the potential association of those serum autoantibodies and NETs in patients with severe carotid stenosis.

## 2. Results

### 2.1. Neutrophils Differentially Accumulate in the Upstream and Downstream Sites of Human Carotid Plaque

Previous work from our group has shown the differential composition of upstream versus downstream segments of atherosclerotic lesions [4]. Here, we have re-analysed carotid plaque composition for the 36 patients included in this sub-study. Demographics and medications of patients involved in this study are described in Table 1. Confirming our previous work, downstream segments/regions of plaques presented a higher accumulation of macrophages and decreased numbers of vascular smooth muscle cells (VSMCs) when compared with upstream plaque segments (Figure 1C,G, respectively). In addition, downstream plaque segments presented a decrease in total collagen and in collagen fractions I and III, and higher expression of MMP-9 as compared to upstream segments (Figure 1D–F,H, respectively). Of importance, the accumulation of neutrophils was significantly elevated in downstream regions of plaques as compared to upstream segments (Figure 1I), and the presence of neutrophils in the downstream segment of the plaque was associated with components of plaque instability, as the number of neutrophils correlated positively with lipid accumulation and negatively with the number of VSMCs and total collagen content (Figure 1J–L, respectively).

### 2.2. Differential Expression of NET Components in Upstream Versus Downstream Segments of Human Carotid Plaques

In different pathological conditions, the expression of NE and H3Cit are associated with the process of NET extrusion by neutrophils [26,27,28,29]. In the present study, immunohistochemistry analysis revealed an increased expression of both NE and H3Cit as NET elements in downstream regions of plaque segments compared with upstream regions (Figure 2A–D). High quality images of whole tissue section have been acquired by using the Axioscan Z1 Slide scanner microscope (Figure 2A,C; top panels). Purple positive signals for NE and H3Cit antibodies can be observed in higher magnification representative images (Figure 2A,C; middle and bottom panels). Immunofluorescent analysis, followed by confocal microscopy, revealed that the intensity of the H3Cit signal was stronger in downstream regions of carotid plaques while the level of H3Cit and neutrophil colocalization was significantly lower in the downstream plaque regions (Figure 3A–C). The most likely explanation for the diminished colocalization of H3Cit and neutrophils in the downstream carotid segments is that H3Cit was already extruded from the neutrophils in the extracellular space. Indeed, H3Cit signal outside of neutrophils was more pronounced in segments from downstream regions than upstream ones (Figure 4A,B). The negative correlation between the H3Cit signal outside neutrophils and the percentage of H3Cit colocalization in neutrophils reinforces our findings and may indicate an alteration in the function of neutrophils present in downstream regions of plaques (Figure 4C). As a final measure of neutrophil functionality, we developed a vortex distance calculation of the H3Cit signal from neutrophils in all possible directions (Appendix A), and we found that the maximum distance was significantly higher in downstream plaque segments compared with upstream segments (Figure 4D).

### 2.3. Pattern of H3Cit Expression in Anti-ApoA-1 Serum Positive Patients

A growing number of clinical studies have suggested that high levels of serum anti-apoA-1 IgG are associated with worse cardiovascular outcomes, including exacerbation of pro-inflammatory biomarkers and features of plaque vulnerability. Indeed, a previous study from our group including the entire cohort showed a significant increase in neutrophil accumulation in downstream segments of human carotid plaques of anti-ApoA-1 IgG serum positive versus negative patients [24]. For this reason, we decided to evaluate whether there was a correlation between NET markers and the serum levels of anti-ApoA-1 IgG of patients included in this sub-study of 36 patients. Our results showed that the serum levels of anti-ApoA-1, estimated as the anti-ApoA-1 index, significantly correlate with the level (%) of the H3Cit signal (Figure 5A), but not with the level (%) of the NE signal (data not shown) in plaques from downstream blood flow portions. Moreover, downstream the blood flow segments of plaques there was a negative correlation between serum anti-ApoA1 index and the extent (%) of H3Cit colocalization in neutrophils, suggesting a more pronounced extrusion of this NET element from neutrophils (Figure 5B). Indeed, anti-ApoA-1 serum positive patients presented a higher signal for H3Cit outside of neutrophils and an increased H3Cit maximal distance from CD66b-positive cells when compared to serum negative patients (Figure 5C–E), suggesting a potential alteration in neutrophil functionality in anti-ApoA1 serum positive patients.

## 3. Discussion

Downstream regions of human carotid plaques have been shown to present features of a more vulnerable phenotype, as compared to areas upstream to the blood flow. In the present study, we showed that not only neutrophil content is different within these two distinct regions, but also their functionality. In line with our data, in a large human cohort enrolling 355 patients, the neutrophil count in carotid arteries was strongly associated with the histopathologic features of rupture-prone atherosclerotic lesions [3]. Neutrophil accumulation was elevated in plaques presenting a large lipid core, high macrophage numbers, a low collagen content, and a small number of VSMC. In addition, previous work from our group showed that the number of neutrophils significantly increases in the carotid plaques of symptomatic patients, defined as those patients whose first episode of ipsilateral stroke occurred 10 to 30 days before endarterectomy, compared to asymptomatic control patients, who presented severe stenosis but no history of ischemic symptoms [4]. Altogether, the above results suggest that even though neutrophils are present in low numbers within atherosclerotic plaques, they can represent an important contributor to plaque destabilisation.

In experimental atherosclerosis, wall shear stress can also determine plaque size and composition. In an elegant model of shear stress-induced atherogenesis, Cheng et al. demonstrated in Apoe^−/−^ mice that low shear stress combined with high cholesterol diet induces the formation of very large plaques presenting a pro-inflammatory and unstable phenotype [30]. By using this model, we previously showed an increased number of neutrophils in plaques developed in low shear stress areas compared to areas exposed to oscillatory flow [31]. In an Apoe^−/−^ mice model of healing and re-endothelization induced by electrical injury, an additional 6 h of flow perturbation induced by the insertion of a perivascular shear stress modifier promoted neutrophil recruitment and thrombus formation in the carotid artery [32]. In the present study, we showed that in human carotid arteries a differential accumulation of neutrophils occurs in the upstream and downstream regions of the plaque. Neutrophil accumulation was enhanced in downstream segments of plaques, and the number of neutrophils was positively correlated with lipid content and negatively correlated with the number of VSMCs and collagen content, underlining the role of neutrophils in plaque instability.

In the present study, segments of plaques in downstream regions presented a higher positive signal for NE and H3Cit, two well-established markers of NETosis. The role of NE in regulating NET release has been elegantly demonstrated in human neutrophils cultured in vitro [33]. However, deletion of NE had only a small effect in reducing NET formation in ionomycin-stimulated mice blood neutrophils, with no effect in PMA-stimulated cells [34]. Moreover, the contribution of NE to atherogenesis is still controversial. Two independent studies showed that Apoe/protein 3/NE triple knockout mice subjected to 4 and 8 weeks of high-fat diet, respectively, exhibited reduced atherosclerotic lesion size compared to Apoe^−/−^ mice [12,35]. Conversely, Wen G. showed that genetic deletion and pharmacological inhibition of NE reduced plaque size in Apoe^−/−^ mice. NET degradation by repetitive DNase injection also showed controversial results in affecting lesion formation in Apoe^−/−^ mice [12,36]. Finally, as NETosis is not the only source of cfDNA, it has been suggested that a combination of surrogate markers for NETs should be assessed simultaneously to accurately determine NETosis [37]. Franck et al. used the expression of NE and H4Cit to identify NETosis in human ruptured and erosion-prone carotid plaques, although for the left coronary arteries, only the expression of H4Cit was associated with NETs in eroded endothelium [38]. In the present study, the expression of both NE and H3Cit, as detected by classical immunohistochemistry, was increased in regions downstream versus upstream of carotid lesions. Further immunofluorescence analysis revealed that the intensity of H3Cit was greater in downstream regions of plaques and preferentially localized in the extracellular space, outside of neutrophils. The maximum distance between H3Cit signal and neutrophils, extrapolated from vortex distance calculation, revealed an increase in the maximum distance between H3Cit NET elements and neutrophils in downstream regions of plaques, suggesting a NET extrusion by neutrophils.

NETs produced by neutrophils have been shown to enhance local inflammation, inducing ROS release [12], SMCs lysis [39] and IL-1β production by macrophages [40], as well as thrombotic and haemorrhagic complications of coronary atherosclerosis [38,41,42]. Pertiwi et al. used MPO, H3Cit and PAD4 antibodies to characterize NETs in human coronary thrombosed lesions and in lesions with intraplaque haemorrhage (IPH) [15]. Of note, NETs elements were present in haemorrhage areas, but were also abundant in adventitia. Interestingly, their experiments revealed a predominance of NETs in neutrophils present in IPH of on-going haemorrhages, rather than in recent or aged haemorrhages. This is very relevant, as IPH may contribute to plaque vulnerability, rendering the lesion susceptible to rupture and, therefore, substantially increasing the risks for cardiovascular complications.

Interestingly, Montecucco et al. showed that mRNA expression of inflammatory chemokines and cytokines, such as CCL2, CCL3, CCL4, and CX3CL1 is increased in regions downstream to human carotid plaques [4]. Further studies are needed to investigate the interaction between NETs and human plaque inflammation. In atherosclerotic lesions, the pro-oxidant and pro-inflammatory environment can promote chemical post-translational modifications in ApoA-1, such as nitration and oxidation, leading to a dysfunctionality of the protein. Montecucco et al. previously showed that for patients positive for serum anti-Apo A-1 IgG, the total number of blood neutrophils tended to increase in the downstream side of the plaque, compared with serum negative patients [24]. In the present study, we found that the expression of H3Cit in downstream plaques segments was positively correlated with serum anti-ApoA-1 index. Furthermore, H3Cit signal outside of neutrophils and the maximum distance of this NET element from neutrophils were enhanced in anti-ApoA-1 IgG serum positive versus serum negative patients. In human neutrophil culture, anti-ApoA-1 IgG increased the migration of neutrophils towards intraplaque chemoattractants, such as CXCL8 and TNF-α [24]. In macrophage culture, anti-ApoA-1 IgG was able to induce a dose-dependent increase in pro-inflammatory markers, such as IL-6 and TNF-α, as well as the translocation of NFκB to the nucleus [43]. Of note, this in vitro pro-inflammatory effect of anti-ApoA-1 IgG was mediated by the TLR2/TLR4/CD14 complex [14,21,44]. Previous work has highlighted the fact that anti-ApoA-1 IgG passive immunization of Apoe^−/−^ mice increases the number of neutrophils in aortic sinus plaques and that this effect is abrogated in both ApoE/TLR2 and ApoE/TLR4 double knockout mice [25]. TLR4 was shown to mediate superoxide-induced NET formation in vitro [45] and in a mouse model of warm ischemia/reperfusion liver injury [46]. Further in vitro studies are needed to verify whether anti-ApoA-1 IgG can directly induce NET formation and also how TLRs contribute to this process.

Taken together, the present work demonstrated the differential expression of NET elements in the upstream and downstream portions of human carotid plaques. This confirmed the concept that human atherosclerotic plaques are highly heterogeneous. Moreover, their capacity to form NETs might be influenced by serum anti-ApoA-1 IgG in patients with severe carotid stenosis, suggesting a new pro-atherosclerotic function for those auto-antibodies in human atherogenesis.

## 4. Methods and Materials

### 4.1. Human Samples

In this ancillary study derived from a princeps study (14), we used patient data related to the histological characterization of internal carotid biopsies from upstream and downstream regions of plaques. As some of the cohort samples had already been processed and published, in the present sub-study we used 36 leftover samples to characterize plaque composition and quantify by immunohistochemistry two NETosis markers, NE and citrullinated histone 3 (H3Cit). For the immunofluorescence protocol followed by confocal images, we used 16 randomly selected upstream and downstream carotid plaque segments. Patients underwent carotid endarterectomy (CEA) for extracranial high-grade internal carotid stenosis (>70% luminal narrowing) according to the recommendations of the North American Symptomatic Carotid Endarterectomy Trial (NASCET) [47], the Asymptomatic Carotid Surgery Trial (ACST) [48] and the European Carotid Surgery Trial (ECST) [49]. The study was approved by the Medical Ethics Committee of San Martino Hospital in Genoa (Italy) (Approval number90/09; 29 July 2009) and participants provided written informed consent. The study was conducted in compliance with the Declaration of Helsinki.

### 4.2. Study Design

Shortly after surgical excision, the human carotid plaque specimens were immediately transferred at 4 °C to the laboratory for processing. All atherosclerotic plaques were cut perpendicular to the long axis through the point of maximum stenosis to obtain both an upstream and a downstream portion (Figure 1A). These two portions have been shown to be exposed to distinct shear stress conditions, influencing plaque composition [4].

### 4.3. Immunostaining and Immunohistochemistry of Endarterectomy Specimens

Upstream and downstream carotid plaque segments embedded in OCT were serially cut into 7 μm sections, and eight sections per each portion were separated by 105 μm from each other. Randomly selected cryosections slides were fixed in cold acetone for 5 min; washed with 1× PBS for 5 min; permeabilized with Triton X 0.1% for 10 min; washed with 1× PBS; and incubated with blocking solution, consisting of 5% BSA in PBS, for 30 min. Endarterectomy specimens were incubated with the following primary antibodies: mouse anti-neutrophil elastase (dilution of 1:100, cat. M0752, DAKO, Agilent Technologies, Santa Clara, CA, USA), rabbit anti-H3Cit (dilution of 1:50, cat. ab5103, Abcam, Cambridge, UK) and rat anti-human CD66b (1:100, cat. IM0166, Beckman Coulter, Switzerland). Primary antibodies were prepared in blocking solution and incubated overnight at 4 °C. After 3× washes with 1× PBS, samples were incubated for 1 h with the following secondary antibodies: Cy3 goat anti-mouse IgG (Jackson Research), Alexa Fluor 594 donkey anti-rat IgG and Alexa Fluor 647 donkey anti-rabbit IgG (Invitrogen) at 1:500 dilution in blocking solution. After 3× washes with 1× PBS, samples were mounted with Vectashield DAPI-containing medium. Immunostaining images were captured by LSM800 Airyscan confocal microscope and analysed with Imaris 9.5, Huygens Essential 19.04. By using the objective of 20×, we tracked neutrophil positive areas, based on CD66b positive staining. Then, in the 40× objective we set up the signal threshold for each antibody, which later on was maintained constant among different sections and groups. For each staining, it captured about three to four images per tissue section, two sections per slide, having 105 μm from each other. By using Imaris software, for each image we measured the total H3Cit signal intensity (signal above threshold level that was set the same for all images captured) and H3Cit signal that colocalized with CD66b. Thus, H3Cit signal outside neutrophils were considered as the difference between total H3Cit signal and H3Cir signal that colocalized with CD66b. Estimation of H3Cit maximal distance to neutrophil was also performed with Imaris software and it is schematically explained in Appendix A. Immunohistochemistry of human specimens was performed similar to the immunofluorescent protocol, however, using VECTOT RED SK-5100 revelation kit (Vector Laboratories, Inc., Burlingame, CA, USA) and the following secondary antibodies: biotinylated anti-mouse IgG and biotinylated anti-rabbit IgG (Vector Laboratories, Inc., Burlingame, CA, USA). NE and H3Cit immunohistochemistry images, represented in Figure 2A,C, were captured by Axioscan Z1 Slide scanner microscope. Prior to scanning with a bright field light, the whole plaque area of each tissue section was manually delineated. After obtaining a high-resolution image, we used Definiens Developer 2.7, Definiens Inc., Carlsbad, CA, USA, delineate the first region of interest (ROI 1), which corresponded to the entire plaque area, and to set up the minimum threshold of signal intensity for each antibody (NE and H3Cit), represented by purple colour positive staining. The sum of all positive signal areas in each section was referred to as ROI 2. The ratio of ROI 2/ROI 1 was expressed as percent of positive signal/plaque area and the final value corresponds to the average of two tissue sections, which were 105 μm from each other. The ratio of ROI 2/ROI 1 was expressed as percent of positive signal/plaque area, and final values correspond to the average of two tissue sections, which were 105 μm from each other.

### 4.4. Assessment of Plaque Vulnerability

Frozen 7 µm cryosections were stained by immunohistochemistry and classic histological coloration techniques as previously described [4]. The following plaque components were evaluated: smooth muscle cells (anti-α-SMA antibody), macrophages (anti-CD68 antibody), collagen total, type I and III (Picrus Sirius red staining, evaluated by polychromatic light and polarized light illumination; respectively), lipid content (Oil-Red Sudan IV), and anti-MMP-9. Images were taken by microscopy and analysed with MetaMorph 7.10, Molecular Devices, LLC. San Jose, CA, USA. Each component was measured and quantified in a given vessel area.

### 4.5. Blood Samples and Determination of Human Auto-Antibodies anti-ApoA-1 by ELISA

Blood samples were obtained by peripheral venipuncture, the day before the endarterectomy. The anti-ApoA-1 IgG index in serum was measured as previously described [22]. Briefly, Maxi-Sorb plates (Nunc) were coated with purified, human-derived delipidated ApoA-1 (20 μg/mL; 50 μL/well) for 1 h (h), at 37 °C. After 3× wash with PBS, all wells were blocked for 1 h, with blocking solution (2% BSA in PBS, 100 μL/well), at 37 °C. Samples were diluted 1:50 in blocking solution and incubated for 60 min. In order to assess individual non-specific binding, additional patient samples, at the same dilution, were also added to an uncoated well. After six washes, 50 μL/well of signal antibody (alkaline phosphatase-conjugated anti-human IgG; Sigma-Aldrich, St. Louis, MO, USA) diluted 1:1000 in blocking solution was incubated for 1 h, at 37 °C. After six more washes (150 μL/well) with blocking solution, the phosphatase substrate *p*-nitrophenyl phosphate disodium dissolved in diethanolamine buffer (pH 9.8) (50 μL/well; Sigma-Aldrich St. Louis, MO, USA) was added. After 20 min of incubation at 37 °C, absorbance at 405 nm (OD405 nm) was determined in duplicates (VersaMax, Molecular Devices, Sunnyvale, CA, USA). For each sample, the corresponding non-specific binding value was subtracted from the mean absorbance value. The positivity cut-off was predefined as previously validated and set (OD value of 0.6 and 37% of the positive control value) [19].

### 4.6. Statistical Analysis

The comparisons between upstream and downstream portions of carotid plaques were performed using the Wilcoxon signed rank test. The comparison between serum anti-apoA-1 IgG positive and negative were performed using Mann–Whitney non-parametric test. Spearman’s rank correlation coefficients were used to assess correlations between variables. All data analysis were performed with Statistica TM software (StatSoft, Tulsa, OK, USA).

## Figures and Tables

**Figure 1 ijms-21-07721-f001:**
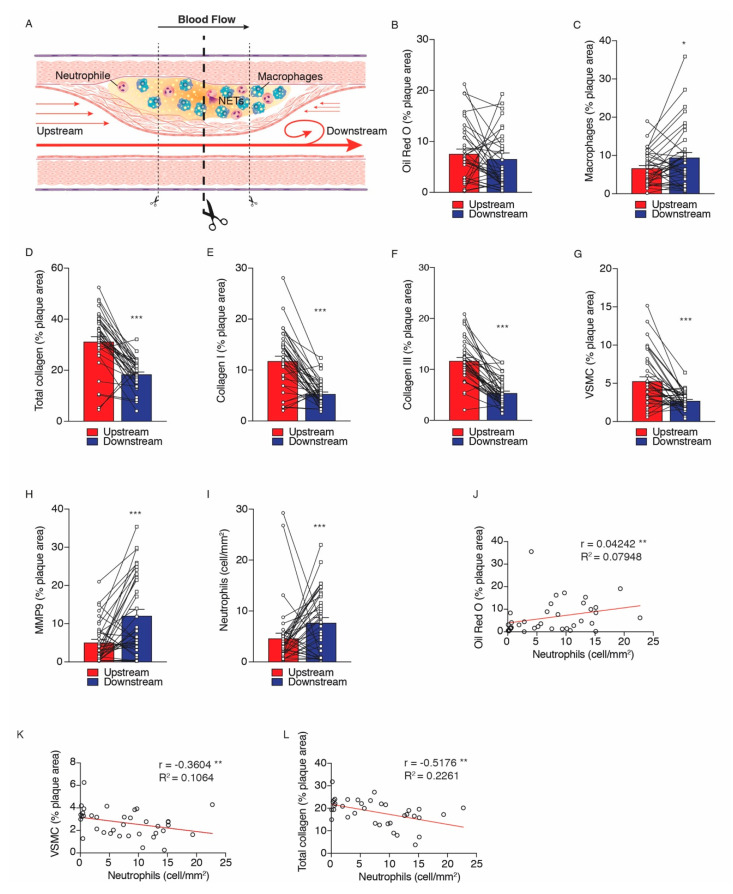
Distinct atherosclerotic plaque composition of upstream versus downstream regions of human carotid artery plaques. (**A**) Representative scheme of sample collection of human carotid plaques located upstream and downstream to the blood flow regions. Bar graph represents the average of immunohistochemistry signal of (**B**) Oil Red O, (**C**) macrophage, (**D**) total collagen, (**E**) collagen I, (**F**) collagen III, (**G**) VSMC, (**H**) MMP-9, and (**I**) neutrophil upstream (red bars) and downstream (blue bars) to the blood flow plaque regions. Spearman’s rank correlation coefficients between Oil Red O (**J**), VSMC (**K**), total collagen (**L**), and neutrophils. Data were expressed as mean ± SEM. *n* = 36 patients for each atherosclerotic region. * *p* < 0.05, ** *p* < 0.01 and *** *p* < 0.001 Wilcoxon Matched Pairs test.

**Figure 2 ijms-21-07721-f002:**
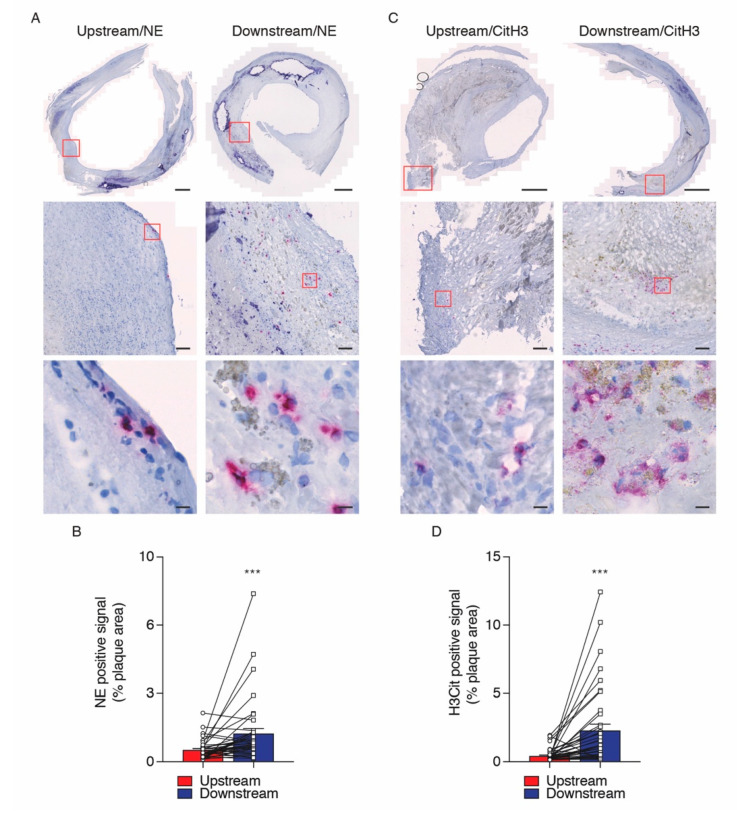
Expression of neutrophil elastase (NE) and citrullinated histone-3 (H3Cit) on upstream versus downstream blood flow plaque regions of human carotid artery. Representative microphotographs of NE (**A**) and H3Cit (**C**) positive signals for whole plaque area (top panels) and two distinct higher magnification images to show specific purple signals for antibody detection of NE and H3Cit (middle and bottom panels). Quantification representing the bar graph of the average immunohistochemistry signal of % NE and H3Cit (**B**,**D**) on upstream (red bars) and downstream (blue bars) total plaque regions of human carotid artery. Data were expressed as mean ± SEM. *n* = 36 patients for each atherosclerotic region. *** *p* < 0.0001 Wilcoxon Matched Pairs test. Scale bar for A and C: 1 mm (top panels) and red frames correspond to area selection for larger magnification images of 100 μm (middle panels) and 10 μm (bottom panels).

**Figure 3 ijms-21-07721-f003:**
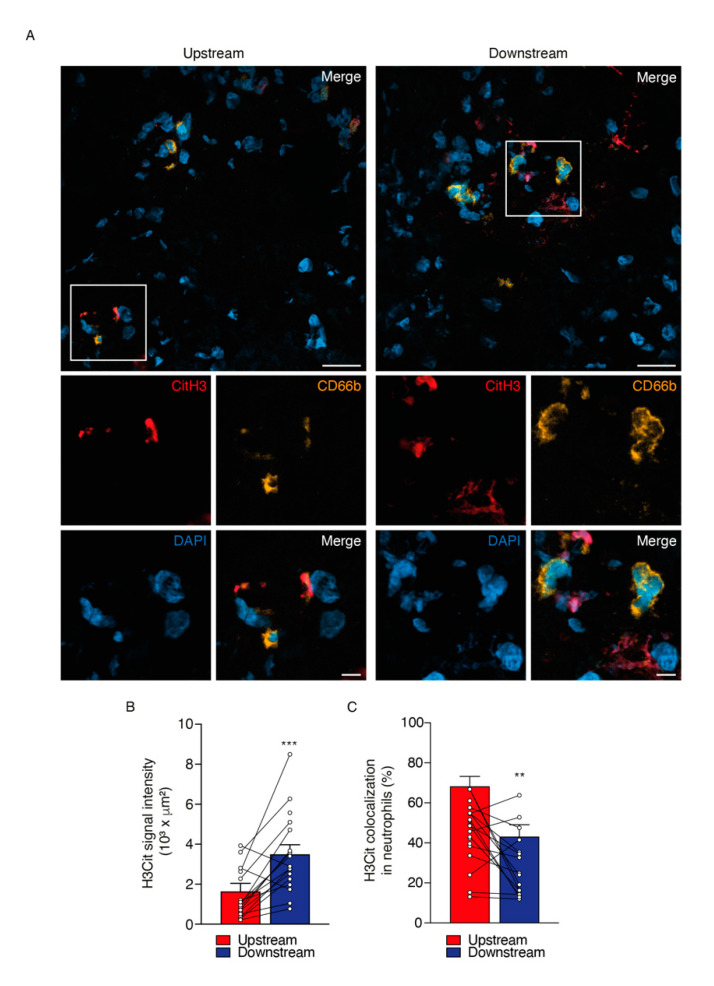
Citrullinated histone-3 (H3Cit) signal intensity and cellular localization on upstream versus downstream plaque regions of human carotid artery. (**A**) Representative microphotographs of immunofluorescence signal for H3Cit (red panels) and its colocalization with neutrophil (CD66b, yellow) and nuclei (DAPI) on upstream (left panels) and downstream (right panels) plaque regions of human carotid artery. Respective quantification representing bar graph average of H3Cit pixel intensity (**B**) and % of colocalization with neutrophil cells (**C**) upstream (red bars) and downstream (blue bars) to the blood flow of plaque regions. Analysis of immunofluorescence images was performed in Imaris software and expressed as mean ± SEM. Scale bar for A: 20 μm (top panels) and white frames in (**A**) represent area selection for larger magnification images of 5 μm (bottom panels). *n* = 16 patients for each atherosclerotic region. *** *p* < 0.001 and ** *p* < 0.01 Wilcoxon Matched Pairs test.

**Figure 4 ijms-21-07721-f004:**
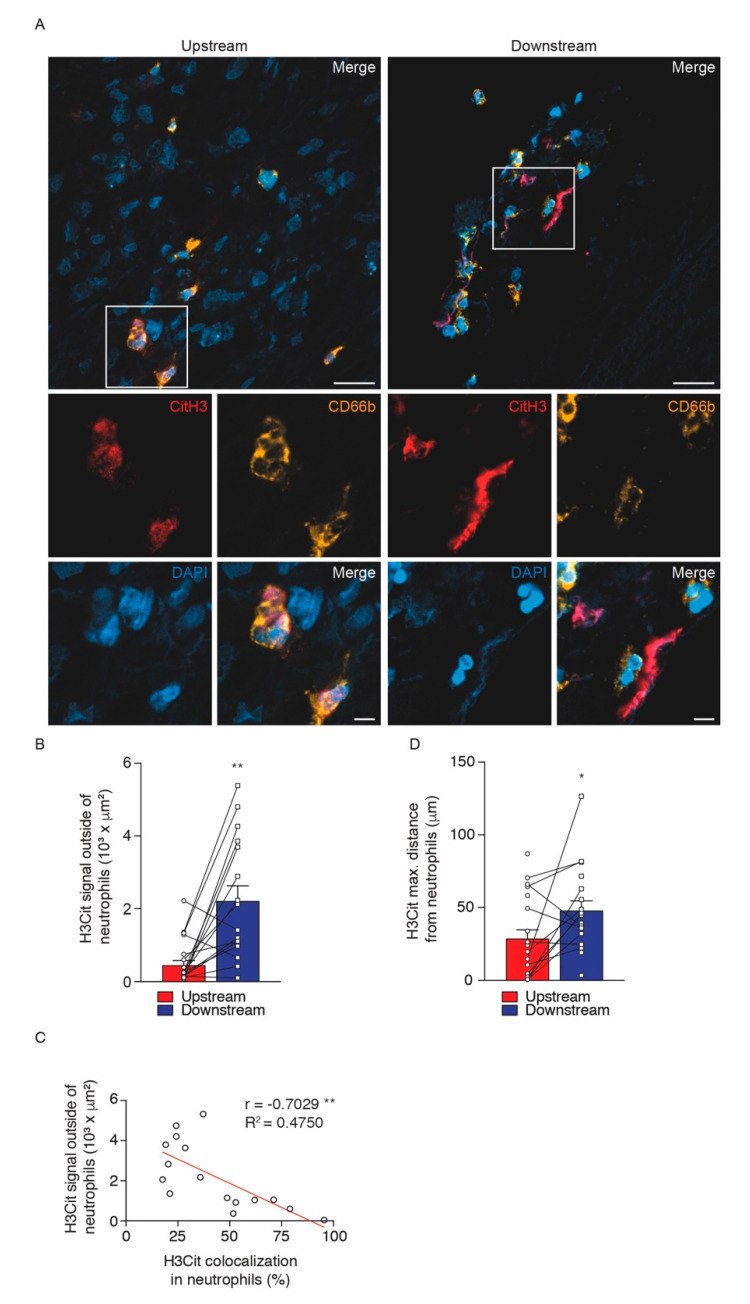
Pattern of Citrullinated histone-3 (H3Cit) signal distribution in upstream versus downstream the blood flow plaque regions of human carotid artery. (**A**) Representative microphotographs of immunofluorescence signal of H3Cit outside of the neutrophil cells on upstream (left panels) and downstream (right panels) plaque regions of human carotid artery. (**B**) Quantification representing bar graph average of H3Cit signal outside the neutrophils and (**C**) Spearman’s rank correlation coefficients between H3Cit signal outside of neutrophils and H3Cit colocalization in neutrophils. (**D**) Maximum distance of H3Cit signal from neutrophil cells in upstream and downstream plaque to the blood flow regions. Vortex distance analysis from immunofluorescence images was quantified in Imaris software and expressed as mean ± SEM. Scale bar for A: 20 μm (top panels) and white frames in A represent area selection for larger magnification images of 5 μm (bottom panels). *n* = 16 patients for each atherosclerotic region. ** *p* < 0.01 and * *p* < 0.05 Wilcoxon Matched Pairs test.

**Figure 5 ijms-21-07721-f005:**
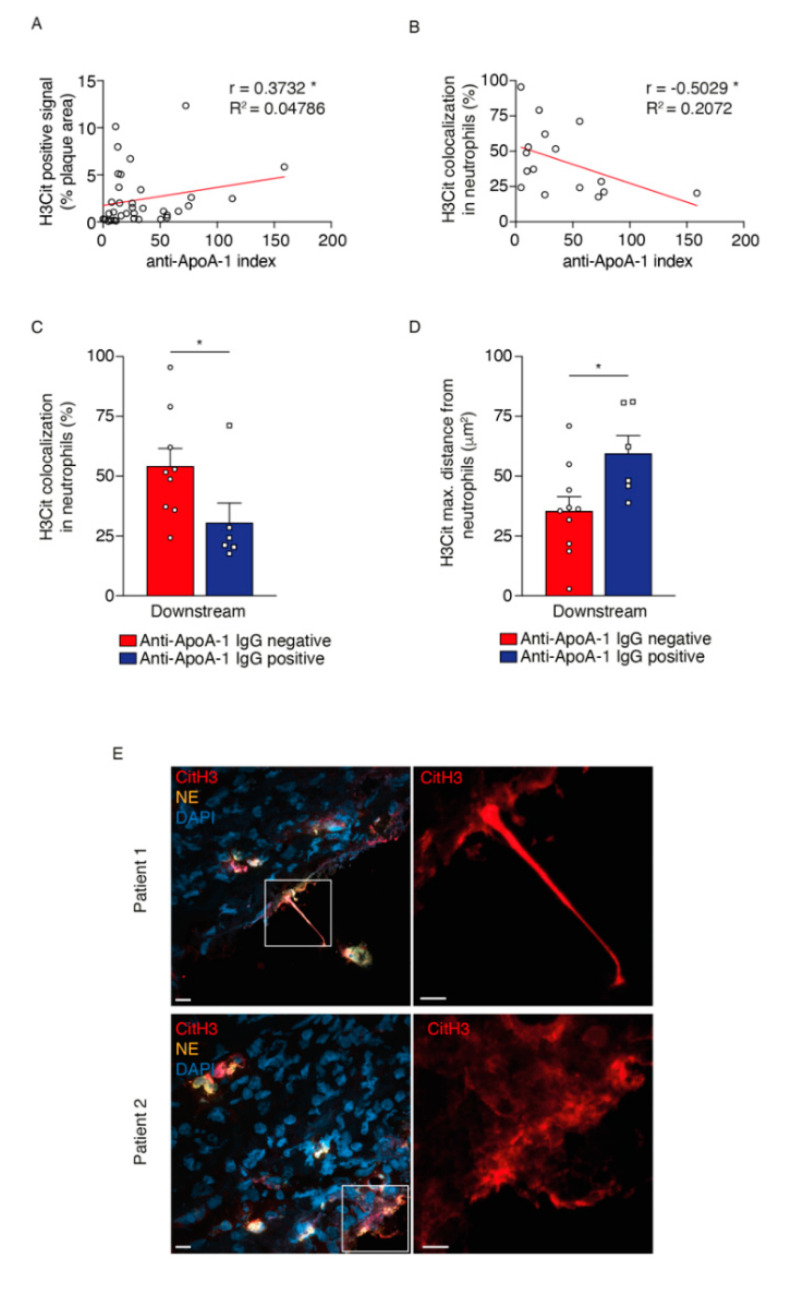
Distinct pattern of citrullinated histone-3 (H3Cit) expression in patient serum positive versus negative for anti-apolipoprotein A-1 IgG (ApoA-1). Spearman correlation coefficients between (**A**) H3Cit positive signal, (**B**) H3Cit colocalization in neutrophils and anti-ApoA-1 index. Quantitative average bar graphs for (**C**) H3Cit colocalization in neutrophils and (**D**) H3Cit maximum distance from neutrophil cells on downstream plaque regions of patient serum negative versus positive for anti-ApoA-1 IgG. (**E**) Representative 3-D microphotographs of immunofluorescence signal for neutrophil elastase (NE) (yellow), H3Cit (red) and nuclei (DAPI) on downstream plaque regions of two distinct anti-ApoA-1 serum positive patients. Analysis of immunofluorescence images was performed by Imaris software. Scale bar for C: 10 μm (left and middle panels) and white frames in E represent area selection for larger magnification images 5 μm (right panels). Data were expressed as mean ± SEM. *n* = 6–10 patients. * *p* < 0.05 Mann–Whitney test.

**Table 1 ijms-21-07721-t001:** Clinical characteristics and medications of the overall cohort admission.

	Overall Cohort (*n* = 36)
Demographics	
Age, year (IQR)	70 (66–75)
Males, number (%)	23 (63.9)
Systolic blood pressure, mmHg (IQR)	140 (130–150)
Diastolic blood pressure, mmHg (IQR)	80 (80–85)
Waist circumference, cm (IQR)	90 (80–100)
Carotid stenosis, % (IQR)	80 (75–86)
Current smoking, number (%)	10 (27.8)
Type 2 diabetes, number (%)	6 (16.7)
Hypertension, number (%)	25 (69.4)
Medications	
ARBs, number (%)	18 (50)
ACE inhibitors, number (%)	1 (2.8)
Beta-blockers, number (%)	8 (22.2)
Calcium channel blockers, number (%)	12 (33.3)
Diuretics, number (%)	3 (8.3)
Statins, number (%)	13 (36.1)
Anti-platelets, number (%)	27 (75)
Clopidogrel, number (%)	5 (13.9)
Oral anti-diabetics, number (%)	3 (8.3)
Insulin, number (%)	0

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
