# Peer review of "Anti-Apolipoprotein A-1 IgG Influences Neutrophil Extracellular Trap Content at Distinct Regions of Human Carotid Plaques"

_ijms, 2020, doi:10.3390/ijms21207721_

Round 1

Reviewer 1 Report

This study investigated NET markers upstream and downstream of atherosclerotic plaques in patients. Significant differences between the two sample sites were found and a link to anti-ApoA-1 was made. The manuscript is well written and scientifically sound.

Major comments

Introduction: The paragraph introducing NETs needs to be improved. NETs were first discovered over 15 years ago by Brinkmann et al in 2004. Furthermore ROS dependent and independent mechanisms have been shown. Additionally, there have been numerous reports detailing the negative effects of NETosis in specific conditions.

Figure 2. Why use average signal intensity instead of number/percentage of positive cells? For figures 2, 3 and 4, the results and interpretation would be improved if individual cells were counted instead signal intensity.

Figure 3: What percentage of DAPI positive cells are neutrophils? If this is known, it would be better to replace figure B with the total number of neutrophils.

Figure 4. How did you quantify H3Cit outside of the neutrophil? Can you be sure that the H3Cit is extracellular?

The manuscript would be improved if the authors assesed the influence of anti-ApoA-1 on NET formation in vitro.

Minor comments

Please use either British or American English, e.g.:

Analysed 87, 301 and 312

destabilization line 202.

210, 215. In the present study used twice.

Line 256. Double space between fact that

Reviewer 2 Report

Authors investigated neutrophils accumulation in atherosclerotic plaques and reported neutrophil extracellular traps (NET) elements as potential markers of vulnerability. This paper is interesting, however following point should be described.

In clinical practice, we sometimes encounter the patient with internal carotid artery occlusion due to intraplaque hemorrhage. Authors should mention the relationship between NET and intraplaque hemorrhage.

Reviewer 3 Report

The authors have investigated the presence of NET markers in upstream and downstream regions of carotid artery atherosclerotic plaques. Samples from plaques were analyzed from patients for NET markers NE and H3Cit, with downstream plaques having a more vulnerable phenotype with elevated NET markers and neutrophils. There was also found to be a correlation between increased anti-apolipoprotein A-1 antibodies and increased H3Cit in downstream plaques. The study is interesting and provides important evidence of using NET markers and anti-apolipoprotein A-1 antibodies to possibly predict atherosclerotic plaque vulnerability. It can be improved by addressing the following points both major and minor:

Major Revisions:

  • Table 1 is mentioned in the text as showing patient demographics, but it is not included in the manuscript or supplementary information. Please include the table in the revised version of the manuscript.
  • For Figure 2, was % NE plaque area and % H3Cit plaque area calculated from the middle panels or bottom panels? If the area was calculated from a subset of images of the entire plaque for each patient, the number of images quantified per patient, the size of each analyzed image, and how the images were selected for analysis should be described in detail in the main text. If the entire plaque is analyzed, that should be stated. Similarly, selection of fluorescence images for analysis in Figures 3 and 4 should be explained in the main text with more detail as per Figure 2.

Minor Revisions:

  • *p<0.05 is mentioned but often *** is shown, which is not defined.
  • It could be mentioned in Figure 2’s caption that the specific staining for NE and H3Cit is shown in purple in the images.
